# Rapid On-Site AI-Assisted Grading for Lung Surgery Based on Optical Coherence Tomography

**DOI:** 10.3390/cancers15225388

**Published:** 2023-11-13

**Authors:** Hung-Chang Liu, Miao-Hui Lin, Wei-Chin Chang, Rui-Cheng Zeng, Yi-Min Wang, Chia-Wei Sun

**Affiliations:** 1Section of Thoracic Surgery, Mackay Memorial Hospital, Taipei City 10449, Taiwan; hcliu@mmh.org.tw; 2Intensive Care Unit, Mackay Memorial Hospital, Taipei City 10449, Taiwan; 3Department of Medicine, Mackay Medical College, New Taipei City 25245, Taiwan; 4Department of Optometry, Mackay Junior College of Medicine, Nursing, and Management, Taipei City 11260, Taiwan; 5Biomedical Optical Imaging Lab, Department of Photonics, College of Electrical and Computer Engineering, National Yang Ming Chiao Tung University, Hsinchu City 30010, Taiwan; cinnamoroll.ee05@nycu.edu.tw (M.-H.L.); nycu0850601.ee08@nycu.edu.tw (R.-C.Z.); hn0937@nycu.edu.tw (Y.-M.W.); 6Department of Pathology, Mackay Memorial Hospital, New Taipei City 25160, Taiwan; 221015@h.tmu.edu.tw; 7Department of Pathology, Taipei Medical University Hospital, Taipei City 11030, Taiwan; 8Department of Pathology, School of Medicine, College of Medicine, Taipei Medical University, Taipei City 11030, Taiwan; 9Institute of Biomedical Engineering, College of Electrical and Computer Engineering, National Yang Ming Chiao Tung University, Hsinchu City 30010, Taiwan; 10Medical Device Innovation and Translation Center, National Yang Ming Chiao Tung University, Taipei City 11259, Taiwan

**Keywords:** lung cancer, optical coherence tomography (OCT), tumor grading, interactive human–machine interface (interactive HMI), deep learning (DL)

## Abstract

**Simple Summary:**

In early-stage lung cancer surgery, determining the extent of resection relies on microscopic examination of frozen sections (FSs), especially when the histology is unknown preoperatively. While optical coherence tomography (OCT) holds promise for instant lung cancer diagnosis, grading tumors with OCT remains challenging. Our study proposes an interactive human–machine interface (HMI) that integrates a mobile OCT system, deep learning, and attention mechanisms. The interactive HMI can mark lesion locations on real-time images and perform tumor grading, aiding clinical decisions. In a trial with twelve preoperatively indeterminate adenocarcinoma patients who underwent thoracoscopic resection, the results of the presented HMI system outperformed frozen sections, achieving an 84.9% overall accuracy compared to FSs’ 20%, showcasing the HMI’s potential for rapid diagnostics and improved patient outcomes.

**Abstract:**

The determination of resection extent traditionally relies on the microscopic invasiveness of frozen sections (FSs) and is crucial for surgery of early lung cancer with preoperatively unknown histology. While previous research has shown the value of optical coherence tomography (OCT) for instant lung cancer diagnosis, tumor grading through OCT remains challenging. Therefore, this study proposes an interactive human–machine interface (HMI) that integrates a mobile OCT system, deep learning algorithms, and attention mechanisms. The system is designed to mark the lesion’s location on the image smartly and perform tumor grading in real time, potentially facilitating clinical decision making. Twelve patients with a preoperatively unknown tumor but a final diagnosis of adenocarcinoma underwent thoracoscopic resection, and the artificial intelligence (AI)-designed system mentioned above was used to measure fresh specimens. Results were compared to FSs benchmarked on permanent pathologic reports. Current results show better differentiating power among minimally invasive adenocarcinoma (MIA), invasive adenocarcinoma (IA), and normal tissue, with an overall accuracy of 84.9%, compared to 20% for FSs. Additionally, the sensitivity and specificity, the sensitivity and specificity were 89% and 82.7% for MIA and 94% and 80.6% for IA, respectively. The results suggest that this AI system can potentially produce rapid and efficient diagnoses and ultimately improve patient outcomes.

## 1. Introduction

According to statistics from the American Cancer Society (ACS) in 2023 [1], lung cancer ranks second in incidence among all cancers affecting both men and women, boasting the highest mortality rate. However, most patients are in the advanced stages of treatment due to the difficulty in the early diagnosis of lung cancer due to its discreet symptoms in the past. A retrospective study found that lung cancer mortality correlated with the time interval from diagnosis to the start of treatment significantly [2]. Thus, with the advancement of imaging technology, more early-stage lung cancer with preoperatively indeterminate histology is found, even incidentally [3]. These early-stage and small lung tumors make rapid on-site intraoperative diagnoses crucial, as the histological grade influences surgical strategies [4] and indirectly affects disease survival.

Presently, the gold standard of microscopic diagnosis is the formalin-fixed paraffin-embedded section (FFPE), in which samples are extracted intraoperatively. Despite the microscopic resolution of FFPE enabling a definite diagnosis, FFPE diagnosis takes a couple of days to obtain results and thus is not applicable for instant diagnosis. As a substitute for FFPE, a frozen section (FS) provides acceptable classification power for lung tumors. The FS has been the standard and only intraoperative diagnostic tool for a long time [5]. Unfortunately, owing to the intrinsic limitations of FSs, the accuracy for grading lung cancer presents constraints and is a matter of debate [6,7,8]. A microscopic examination of intact tissue morphology is needed to achieve high differentiation power. However, according to the literature reviewed, frozen sections still have certain risks of misleading surgical procedures with an accuracy of 37~95% [4,7]. Developing an optional or assistant method for intraoperative histologic diagnoses is always required. Optical coherence tomography (OCT) was reported to permit real-time and depth-resolved images with submicron resolution. OCT, being non-contact, non-invasive, and non-radiative entities, is based on the principle of low coherence interferometry. Previous research verified that OCT was a potentially promising tool for assisting lung tumor surgery [9,10]. OCT could distinguish intraoperatively between cancerous and normal tissue on fresh ex vivo specimens [9]. Furthermore, OCT could provide strengths of qualitative and additional quantitative analysis for lung tumors [10]. This OCT technology can create compatible images through instant scanning within seconds to minutes. Furthermore, digitalized images make the differentiated diagnoses of instant images easier and optimized, as the benefits of digitalized data can be processed through artificial intelligence (AI).

The power of AI has recently been reported with outstanding growth. The continuous management of incoming data processed by AI increased sensitivity and decreased mistakes. OCT imaging in combination with AI has also emerged for various types of cancer [10,11,12,13,14,15,16,17,18,19]. For example, in the study of breast cancer, a deep neural network (DNN) was used to perform real-time edge assessment in breast lumpectomy surgery, using AI to identify the edge of breast cancer [10]. One of the more exciting articles uses reverse active learning to diagnose breast cancer OCT images [18]. However, more trials still need to discuss the contributions of this new technology to diagnoses of cancer-related lung tumors. Accordingly, the current study is designed to explore the potential roles of the combination of OCT and AI technology in tumor diagnoses.

## 2. Materials and Methods

This clinical trial for lung tumors did not entail direct human body intervention; all tissue specimens remained confined within the hospital premises. The study was approved by the Institutional Review Board (IRB) of Mackay Memorial Hospital, focusing exclusively on primary lung tumor patients aged 20 to 80, with an explicit exclusion criterion for individuals with metastatic carcinoma or who had previously undergone targeted therapy, systemic radiation therapy, or chemotherapy. The OCT cart was systematically deployed during surgical procedures to practice clinical scenarios in the operating room. Subsequently, ex vivo tissue samples excised from operations were utilized as the studied subjects for this comprehensive research endeavor.

The surgical team promptly intraoperatively provided lesions and normal tissue samples during the initial phase. The normal tissue was at least 20 mm away from the maximal visible area of the lesion. These specimens served as the foundational elements for establishing a comprehensive database. Each piece of tissue was approximately 4 mm × 4 mm × 4 mm and underwent rapid OCT-performed three-dimensional scanning with image reconstruction. The scanning area was 2.4 mm × 2.4 mm. Any specimen could be scanned multiple times at different angles to increase sample size diversity. This process included a data access time of about several minutes. At the same time, the surgeon also sent a small piece of the lesion specimen to the pathology department for an intraoperative FS. It took about 20 min to wait for a preliminary diagnosis by the pathologist. As for the final pathologic report, the definitive benchmarked diagnosis was available about a week later. Regarding the pathologic diagnosis, adenocarcinoma in situ (AIS) is defined as a small adenocarcinoma (≤3 cm) with a pure lepidic growth pattern while lacking any stromal, lymphovascular, pleural, alveolar space invasion or necrosis. Minimally invasive adenocarcinoma (MIA) is defined as an adenocarcinoma (≤3 cm) with a predominant lepidic pattern and ≤5 mm invasive component (such as acinar, papillary, solid, or micropapillary patterns), whereas invasive adenocarcinoma (IA) is defined as an adenocarcinoma with an invasive component measuring >5 mm in its greatest dimension.

As previously published [20], the customized SD-OCT system was employed in this research. Figure 1 illustrates our experimental setup, characterized by a single-mode fiber-based unbalanced Michelson interferometer configuration. The wavelength of the light source emanates from broadband superluminescent diodes (SLDs) with an average output power of 11 mW (cBLMD-S-371-HP2-SM-OI, Superlum, Carrigtohill, Ireland). The laser’s wavelength is centered at 840 nm, with a full width at half maximum (FWHM) of 51.2 nm, thereby attaining a theoretically calculated axial resolution of 6.06 μm. The lateral resolution, defined by the spot size measurement at the focal plane, approximates 10 μm (in air). The A-line scanning rate achieves 20 kHz, and the spectral interference signals are efficiently acquired by a commercially available spectrometer (Cobra-800-880, Wasatch Photonics, Logan, UT, USA). These spectral signals are then seamlessly converted from analog to digital (A/D) format using a personal computer.

In the fiber pathway, the only essential component is a 50/50 coupler (C). The light emitted by the superluminescent diode (SLD) initially traverses through this coupler; after that, the resultant twin beams propagate toward the reference and sample arms. In the reference arm, the beam is collimated utilizing a fiber collimator (FC2) (F280APC-850, Thorlabs Inc., Lafayette, CO, USA) incorporating an adjustable neutral density filter (ND-filter) and achromatic lens (L1) (AC254-030-B-ML, Thorlabs Inc., Lafayette, CO, USA), subsequently encountering reflection by a silver-coated mirror (M). As for the sample arm, the beam traverses through a pair of precisely controlled galvanometric scanning mirrors (G1 and G2) (6220H Series, Cambridge Technology Inc., Peachtree Corners, GA, USA). These beams emanating from the reference and sample arms converge with each other at C, thereby engendering interference signals and introducing them into the spectrometer. The spectrometer comprises a transmission grating and a linear line scan camera featuring 2048 pixels. Ultimately, the data flow to our PC is facilitated through a Camera Link connection. The hardware flowchart of the system was designed using a self-built LabVIEW program. (LabVIEW 2016, National Instruments, Austin, TX, USA). Furthermore, the data acquisition device (DAQ) (USB 6343, National Instruments Inc., Austin, TX, USA) adeptly governs the precise movement of G1 and G2 to generate two-dimensional (2D) or three-dimensional (3D) waveforms. The acquired data are then transmitted to the PC via a Camera Link connection.

The practical frame rate is 20 frames per second (fps). In conditions where shot noise predominates, the theoretically calculated sensitivity of this architecture is 110.3 dB. With an ND-filter attenuation of 50 dB, the measured signal-to-noise ratio (SNR) is 40 dB, indicating a current sensitivity of 90 dB. The scanning range covers 2.4 mm along both axes. G1 carries out 2000 line scans, each lasting 0.1 s, while G2 operates at 0.025 Hz, necessitating 40 s to complete a single C-scan volume. Consequently, 400 B-scan images (2D) were amalgamated to construct a volumetric set (3D). As confirmed by the surgical team, we conducted volumetric measurements at two distinct tissue sites: the lesion and the normal.

Overall processing was performed using the programming language Python v3.8, harnessing the power of CUDA GPU acceleration on a high-performance Windows-based computer boasting 16.0 GB of RAM, an Intel Core i5-7500 CPU, and an NVIDIA GeForce GTX1660 GPU. The flowchart is shown in Figure 2. All raw data first underwent calibration of k linearity and window cropping. Subsequently, to mitigate speckle noise, we generated despeckled images by employing an averaging approach across seven adjacent B-scans after translational registration. These images were then resized and normalized into the size of 128 pixels (depth) × 218 pixels (width), corresponding to an actual scan range of 1.4 mm (depth) × 2.4 mm (width). The values were rescaled from 0 to 1 before importing them into the training convolution neural network. Furthermore, data augmentation was rigorously implemented by incorporating random combinations of width and height shifting, shearing, zooming, and horizontal flipping. Notably, to safeguard the integrity of the morphological features in the OCT images, the parameters governing shearing and zooming were meticulously set at a conservative ratio of 0.1.

The current study’s neural network model structure aligns with our prior publication [21]; it is an attention-mechanism-based ResNet model for classifying brain tumor tissue OCT images, rendering it a pertinent candidate for our lung tumor dataset. In the realm of neural network design, the attention ResNet model comprises 14 layers featuring six optimized residual blocks, as elucidated in Figure 3a. Notably, attention mechanisms have been thoughtfully integrated into the final residual block, incorporating filter sizes of 32 and 64. This augmentation ensures the attention mechanism remains fully engaged, even when capturing rudimentary features. Furthermore, a supplementary attention path, as depicted in Figure 3b, has been introduced. This design permits the attention mechanism to gracefully attenuate during training if it struggles to discern more pertinent features. The neural network effectively captures superior features when the Alpha (α) value exceeds zero. During the training phase, a batch size of 16 images was employed, and the stochastic gradient descent (SGD) optimizer was selected with a learning rate of 0.0001 and a momentum value of 0.9. The activation function used Leaky ReLU, and the value was set to 0.01 to prevent the output and input from showing a linear relationship. Furthermore, the additional L2 regularization (weight 0.05) penalty term was used to avoid the overfitting of the model and make the loss function smoother. Categorical cross-entropy, one of the most commonly used loss functions, served as the criterion to evaluate optimization performance, and the training process concluded when the validation data’s loss function ceased to decrease for 15 consecutive epochs.

T-distributed stochastic neighbor embedding (t-SNE), a non-linear machine learning dimensionality reduction method, can maintain local structure during dimensionality reduction. It was proposed by Laurens van der Maaten and Geoffrey Hinton in 2008 [22]. Gaussian distribution with low-dimensional information through t-distribution and Kullback–Leibler divergence (KLD) calculation are performed for the similarity of two probability density functions and gradient descent to find the best solution. On the other hand, gradient class activation mapping (grad-CAM), an innovation by R. R. Selvaraju [23], is modified as classification decisions. Python v3.8, a cloud-based database saver to back up collected patient data and associated information, and Matplotlib’s mouse-responsive functionalities for unveiling individualized OCT characteristics within the t-SNE image are also applied in the current study.

## 3. Results

Twelve patients with a preoperatively indeterminate tumor underwent thoracoscopic resection. Five patients permanently diagnosed with minimally invasive adenocarcinoma (MIA) and seven with invasive adenocarcinoma (IA) were recruited in this study. The specimens were extracted during the ordinal operations. Any specimen might be scanned multiple times using OCT to increase the data. The recruitment details are listed in Table 1. We divided the data into the training dataset and the testing dataset. All data separations were based on individual patients to generalize the model applicability to unseen OCT images, as shown in Table 2.

The overall accuracy derived from the confusion matrix stands at an impressive 84.9%, with individual class accuracy exceeding 80% for each category, demonstrating an acceptable classification capability. Figure 4 shows the confusion matrix from our model, including the number of pictures (Figure 4a) and normalized probability (Figure 4b). Specifically, for MIA, the sensitivity and specificity are 89% and 82.7%, respectively, while for IA, they are 94% and 80.6%.

Figure 5 shows the trend of changes concerning the epochs. We plot the learning curve in the processing. Figure 5a shows the accuracy of training (red line) and testing (green line) processing. The accuracy of the test data exhibited significant fluctuations initially, stabilizing and converging starting from the 150 epochs. Furthermore, the loss curve of the test data (blue) is slightly higher than that of the training data (yellow). It exhibits a gradual plateauing trend over epochs, highlighting the stability of our model. In order to better understand the model performance, the receiver operating characteristic (ROC) curves of IA and MIA were calculated from the testing data, as shown in Figure 5b,c. The areas under the curves (AUCs) were 0.99 and 0.96, showing excellent differentiation powers for the targeted tumors. The AUC of the ROC curve reflects the comprehensive performance of the model under different thresholds if the model can distinguish between positive and negative classes well and maintain sensitivity and specificity under various thresholds.

Subsequently, we harnessed the capabilities of t-SNE in conjunction with OCT images and grad-CAM to ascertain whether the model authentically fixates on the correct features instead of background information (Figure 6). Within the t-SNE interface, the posteriorly situated lighter-colored data points represent the training dataset, while the anterior darker-colored data points signify the test dataset. Data points of matching hues on the t-SNE image denote the same category, with a t-SNE perplexity parameter set to 35. The t-SNE conclusively reveals that data points of similar colors coalesce into distinct clusters, with only minor instances of overlapping with other categories. This affirms that our model has undergone robust training and can effectively discriminate between the three types. Misclassified data points will be discussed further in subsequent sections.

Moreover, exploring the analysis of grad-CAM heatmaps, we examined the intricate features extracted by our model within each histology-graded distinct category. In IA images, we observe a phenomenon characterized by interrupted attenuation with discontinuous reflection (Figure 6a). This feature is absent in MIA images. Remarkably, MIA images show relatively formless homogeneity (Figure 6b). In the normal tissue images (NOR), the model focuses on the tissue’s superficial regions, which show irregular and dense spots (Figure 6c). However, the attenuation discontinuity in NOR appears randomly. 

In addition to the reasonably good visual results obtained from t-SNE images, we have also implemented an interactive t-SNE interface. When individual data points within the interface are clicked on, the corresponding OCT image data are promptly retrieved alongside a window presenting pertinent patient information and the related OCT image data. In a separate interface, patient details such as name, chart number, gender, sample number, volume number, and the number of B-scans, in addition to the neural network’s classification probabilities and grad-CAM image, are presented. This feature affords us the immediate capability to scrutinize and analyze our dataset. Furthermore, predictions for the respective data points are provided beneath this window, with associated probabilities and visual CAM images. Figure 6 vividly illustrates the interactive outcomes attained by selecting MIA (Figure 7a) and IA (Figure 7b) data points. These images conclusively demonstrate the neural network’s adeptness at delivering precise predictions for both MIA and IA cases, with the CAM images discerningly highlighting the relevant features of interest rather than fixating on extraneous background information.

## 4. Discussion

Both diagnoses and treatments in medicine are crucial in real-world practice. Lungs are an organ of combined solid and luminal structures in the deep body. Specialists in managing lung tumors, especially for minor conditions increasing incredibly slowly in size, are faced with combined procedures of instant diagnostic judgments between benign and malignant lesions and subsequent surgical treatments. Surgery is considered the standard management for early-stage non-small-cell lung cancer, as small and early tumors in the pulmonary parenchyma are hard to access through bronchoscopy for diagnosis [24]. Therefore, rapid on-site accurate diagnoses, as eagerly needed, provide essential data for subsequent surgical strategies and are especially imperative in pulmonary oncology. Currently, definite diagnosis of tumor entities in the microscopic spectrum can only be supplied by pathologic procedures days after the surgery, leaving time-effective issues for improvement. As a result, instant biopsy through FS reports becomes essential to approach histological data; it is performed by the running duty-shifted pathologist, taking slices at one location to determine a diagnosis. However, detailed incorrect results commonly exist in particularly small lesions. 

Compared with FSs, OCT provides continuous slice images with a range of around 2.4 mm × 2.4 mm. Generally, surgeons can distinguish the tumor area with bare eyes. However, the final pathology must still confirm the invasive regions’ existence and detailed content. If the resected specimen for OCT scanning does not contain enough invasive parts, the model may lead to misclassifications.

The evolutionary development of cell biology for lung adenocarcinoma is believed to proceed sequentially through atypical adenomatous hyperplasia (AAH), adenocarcinoma in situ (AIS), MIA, and overt IA [25,26]. Cancer cells sometimes take a long time to grow, even up to several years. Improvements in the health environment and people’s medical vigilance result in more and more clinical cases of early-stage cancer.

According to literature statistics, the five-year survival rate of early-stage cancer with immediate treatment, especially before the MIA stage, can reach 100% [27]. An important issue is the opposite considerations between the resection criteria extents required for adequate tumor clearance and maximal preservation of lung volume to preserve better pulmonary function. Both destruction and protection are considered crucial. Therefore, surgical treatments according to the set reference based on histological grade played an essential role and were required. Generally, the histologic MIA status of lung tumors is considered the cut-off point for the extent of surgical resection. Beyond MIA, limited resection with so-called sublobar resection was adequate and recommended [28]. Above IA, however, extensive resection by lobectomy or more is often needed for sufficient cancer clearance [24,28,29,30,31]. The current study shows better discrimination capability by OCT integrated with AI (OCT-AI) compared to the traditional FSs. In addition, time saving is another strength of the current electromechanical system (EMS). This OCT-AI can provide data management within tens of seconds to a few minutes, while an FS usually needs at least 30 min. Thus, such an efficient AI-integrated EMS would have the potential to be an optional tool for rapid on-site diagnoses of preoperatively indeterminate tumors.

Although the current model achieves an accuracy rate of approximately 80%, mis-classifications still exist, as with other tools. By scrutinizing the features contributing to misclassifications in the CAM images, we can gain insights into the sources of error (Figure 8). Figure 8a illustrates a scenario in which MIA is erroneously classified as IA, while Figure 8b depicts IA being misclassified as MIA. In Figure 8a, the model principally directs attention to areas characterized by discontinuous attenuation (red arrow of Figure 7a), a hallmark feature of IA in OCT images, as mentioned before. In Figure 7b, the model fixates on relatively uniform regions (blue arrow of Figure 8b), resulting in a misclassification as MIA. Notably, there are also structureless areas near the surface (blue arrow of Figure 8a), indicative of MIA features, yet the model simultaneously focuses on the regions of discontinuous attenuation below. Similarly, some dense spots (red arrow of Figure 8b), the feature of IA, exist in Figure 8b. We infer that the simultaneous co-existence of IA and MIA features would lead to misclassifications. However, the mistakes of such conditions are very low, with a probability of around 4% (misclassification of IA) and 11% (misclassification of MIA).

Tumor spread through air spaces (STAS) is prevalent in regular pathologic findings [32]. In the current study, specimens of normal tissue of around 20 mm or at least the size of the lesions are provided for OCT scanning as a calibrated benchmark. Tumor resections of at least 20 mm are commonly recommended for suspected malignancy, which was the previously generally accepted safety margin [33,34]. The tumor margin distance is a primary concern regarding local control. Therefore, the current study chose normal tissue with a 20 mm distance or at least the size of the lesions for evaluation. STAS is challenging to interpret in a frozen section specimen, and better accuracy is needed [35,36]. On the other hand, as shown in the confusion matrix in Figure 4, normal tissue images can sometimes be erroneously categorized as IA. These conditions are observed to present in all categories but are mostly misclassified in IA specimens. Through quantitative comparison, IA and NOR determined by the OCT-AI system show a particular possibility of mixed discrimination. However, the strengths of OCT-AI lie in the capability of calibrated accuracy by self-promotion through extensive data collection. STAS in lung cancer has been reported to have numerous associations with poor survival [29,36]. Thus, from the concept of treatment effectiveness, wide excision by lobectomy would be suggested if the OCT-AI system reads specimens to be IA of small tumors.

In this era with many pieces of AI-based software, there are some considered suitable for applicable approaches for the current study, such as support vector machines (SVMs) and artificial neural networks (ANNs), including feedforward neural networks (FNNs), recurrent neural networks (RNNs), and convolutional neural networks (CNNs). In this study, a CNN’s convolutional computation was chosen, utilizing convolutional layers to learn features from data automatically [21]. We gave up using SVMs, ANNs, FNNs, and RNNs because there were defects of requiring manual feature extraction tasks and a lack of automatic learning of features from data. An attention-based ResNet model combined with a CNN was selected owing to the strengths of good classification performance for lung cancer. The advantages lay in that this combination reduced the need for extensive domain knowledge and experience and was less demanding in terms of raw data, as well as parameter tuning for good performance. CNNs are typically used in tasks related to computer vision, image classification, and image recognition and are particularly suitable for managing spatial data.

The AI training process is like a complex matrix, and it is difficult to understand the learning details and interpretation basis intuitively. Therefore, it is necessary to use visualization tools to perform dimensional reduction. Standard methods are principal component analysis (PCA) and t-distributed stochastic neighbor embedding (t-SNE). PCA is a linear dimensionality reduction technique that may not capture complex, non-linear relationships in the data as effectively as t-SNE. Others, such as locally linear embedding (LLE), aim to preserve local linear relationships between data points. Isomap is suitable for data with intrinsic non-linear structures but may be sensitive to noise and require careful parameter tuning. In this study, t-SNE was employed for model visualization. 

A module with non-linear machine learning dimensionality reduction, t-SNE, is used. Simultaneously, a modified method designed by Laurens van der Maaten and Geoffrey Hinton in 2008 was proposed [22]. The main idea is to approximate high-dimensional information through Gaussian distribution and low-dimensional information through t-distribution. Kullback–Leibler divergence (KLD) calculation is performed for the similarity of two probability density functions and gradient descent to find the best solution. In the current research, t-SNE was judiciously applied to scrutinize the distributions of data emanating from diverse samples and elucidate their inherent properties. On the other hand, gradient class activation mapping (grad-CAM) was systematically employed for comprehensive model evaluation. Grad-CAM, an innovation pioneered by R. R. Selvaraju [23], facilitates the mapping of regions within an image that significantly influences the model’s classification decisions. This activation map is derived from gradient calculations to the output concerning a given input image. In our specific application, grad-CAM was harnessed to visualize the salient features prioritized by the trained model. 

Validating the model’s training through result visualization is of paramount importance. However, even with excellent training outcomes, the practical utility for clinical healthcare professionals often hinges on the involvement of engineers. Therefore, designing an interactive human–machine interface (HMI) aims to make the user interface easily understood and provide clinical personnel with real-time patient information, an assessment of symptom severity, and an approximate lesion location [37,38,39,40]. An HMI enables clinicians to formulate the quickest surgical strategies, ultimately improving patient outcomes.

The designed processing of the system involved the creation of the interface using Python v3.8. A cloud-based database for backing up collected patient data and associated information, which can be modified and expanded upon in the patient data section, was created. After importing the dataset into the neural network for post-model training, classification probabilities of t-SNE and grad-CAM images were obtained. The Matplotlib plotting library within Python provides an object-oriented application programming interface (API) for embedding graphics into the application interface. Due to Matplotlib’s mouse-responsive functionalities, clicking on individual points within the t-SNE image unveils its OCT images. In a separate interface, patient details such as name, chart number, gender, sample number, volume number, and the number of B-scans, in addition to the neural network’s classification probabilities and grad-CAM image, are presented. This integrated interface is engineered to provide a holistic depiction of the final classification results.

Interactive HMI implementation can provide a friendly and straightforward operating interface on the clinical terminal, whether the classification is correct or incorrect. In addition, quantitative probabilities can provide suggestive data for surgeons with alternative diagnostic guidance. Figure 9 shows two misjudgments on the interactive HMI. The model predicts three probabilities of MIA, IA, and NOR; the final prediction is the largest. Although misclassification, as also exhibited by other diagnostic tools, existed in some cases, the OCT-AI system provided the additional probability data of tumor categories, as displayed on the left side of the interactive HMI, to be considered for decisional judgment by the clinicians. Figure 9a illustrates an NOR from an IA patient erroneously categorized as IA. Probabilities are IA: 66.04% and NOR: 33.96%. Figure 9b indicates the model assignment of reasonably similar possibilities for both IA and MIA. Hypothetically, MIA and IA exist simultaneously. This interface design can empower physicians to access detailed probability information of misclassifications promptly, helping them make well-informed decisions during surgical procedures. 

## 5. Conclusions

This study represents the first real-time clinical investigation of lung cancer tissue using the AI-integrated OCT for grading classification. Our customized neural network model can accurately classify normal tissue, MIA, and IA. Additionally, we have developed an interactive HMI that allows clinicians to assess deep and detailed tumor information, including digital data from OCT scans and, most importantly, visualized images and available predictive probability assessment only by clicking on data points in t-SNE images. Despite the current model’s high accuracy, some misclassifications still exist, indicating spaces left for improvement. In the future, the authors aim to gather more data to enhance the model’s reliability; incorporate features such as in situ cancer, precancerous lesions, and benign conditions; and achieve a more comprehensive and nuanced classification system. In summary, the current proposed method has the potential to enhance traditional pathological examination, improve diagnostic efficiency, and assist patients with ensuring appropriate treatment and better outcomes.

## Figures and Tables

**Figure 1 cancers-15-05388-f001:**
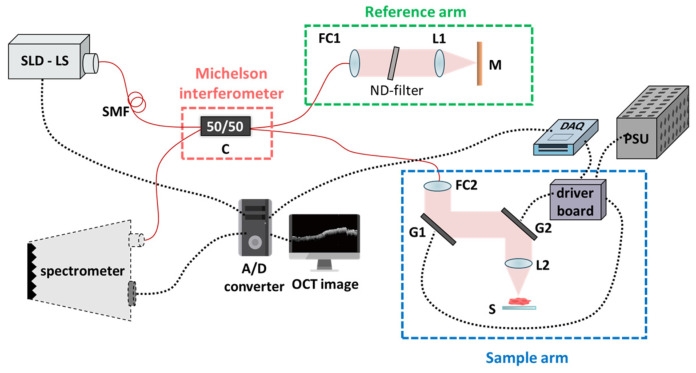
Schematic diagram of the SD-OCT. Red solid lines are the fiber path, black dotted lines are the electric path, and the colored areas are near-infrared in free space. SLD-LS, superluminescent diode light source; SMF, single-mode fiber; C, coupler; FC1 and FC2, fiber collimator; ND-filter, neutral density filter; L1 and L2, achromat lens; M, mirror; S, sample platform; G1 and G2, galvano scanners; DAQ, data acquisition (NI-6343, National Instruments); PSU, power supply; A/D converter, analog-to-digital converter.

**Figure 2 cancers-15-05388-f002:**
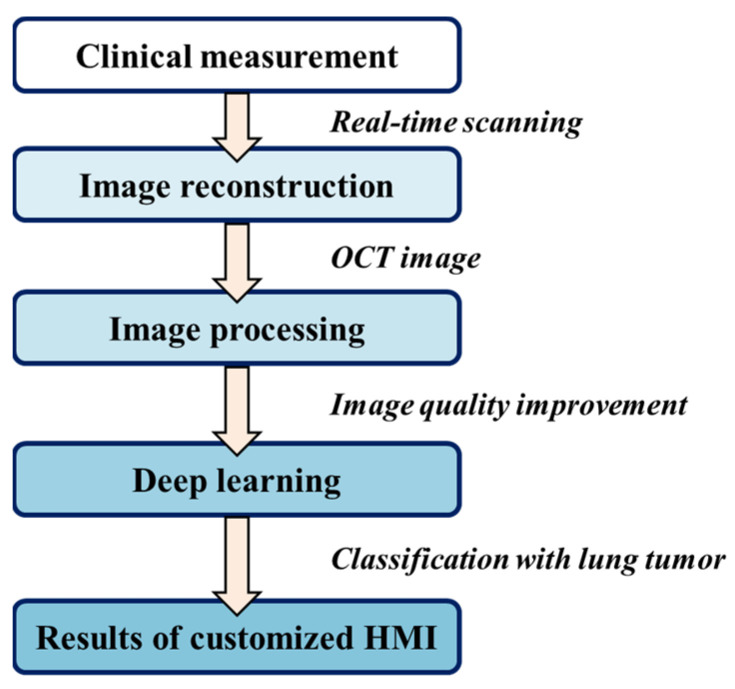
The overall process of the research. The clinically obtained tissues are measured in real time. The data are then reconstructed, pre-processed, and imported into the neural network to obtain the tissue classification results. Finally, the interactive human–machine interface (HMI) is used to verify the correctness of the results.

**Figure 3 cancers-15-05388-f003:**
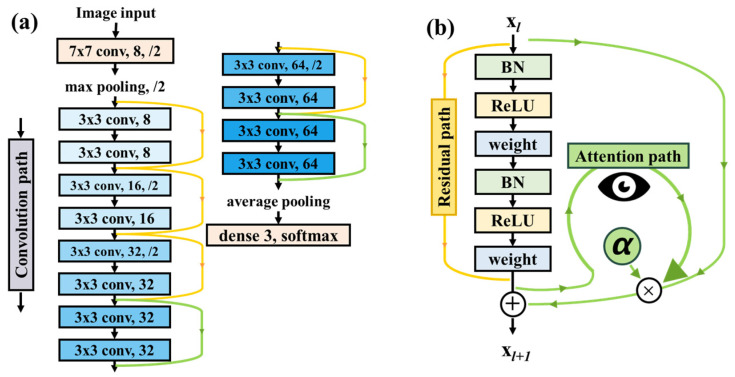
Attention ResNet model architecture diagram [12]. (**a**) Neural network ResNet is designed and (**b**) attention mechanism is added to improve training results. Yellow arrow: residual path. Green arrow: attention path.

**Figure 4 cancers-15-05388-f004:**
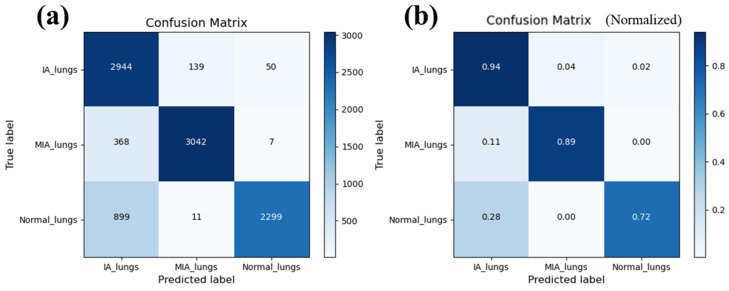
Confusion matrix from testing data of our model. (**a**) is the number of pictures, (**b**) is the normalized probability. The sensitivities and specificities of MIA and IA were 89%, 82.7%, 94%, and 80.6%, leading to an overall accuracy of 84.9%.

**Figure 5 cancers-15-05388-f005:**
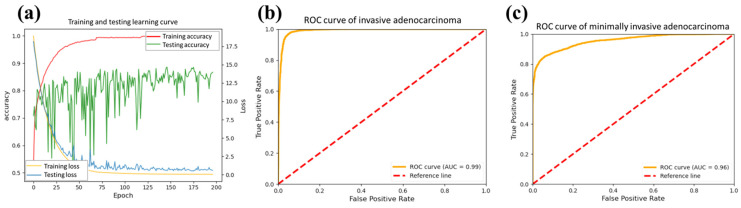
Learning curve and ROC curve. (**a**) is the learning curve of training and testing data, including their accuracy and loss with respect to epochs. The ROC curve of (**b**) invasive adenocarcinoma (IA) and (**c**) minimally invasive adenocarcinoma (MIA). The areas under the curve (AUCs) are 0.99 and 0.96, respectively.

**Figure 6 cancers-15-05388-f006:**
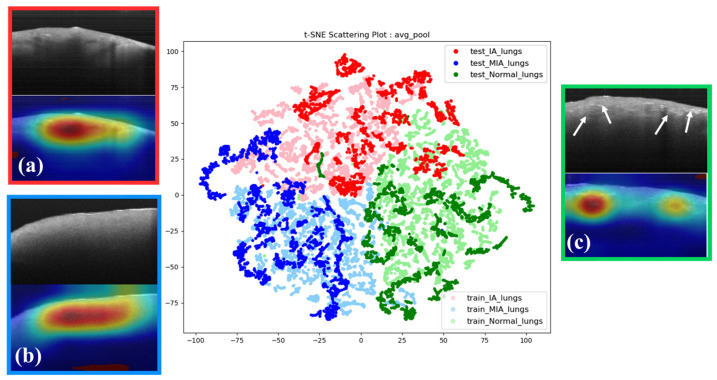
Model performance visualization t-SNE diagram: (**a**) IA, (**b**) MIA, and (**c**) NOR OCT images of lung tissue observed from a subjective point of view (top) paired with the grad-CAM diagrams (bottom) that the model focuses on. White arrows show that NOR tissue has an abundance of dense light spots.

**Figure 7 cancers-15-05388-f007:**
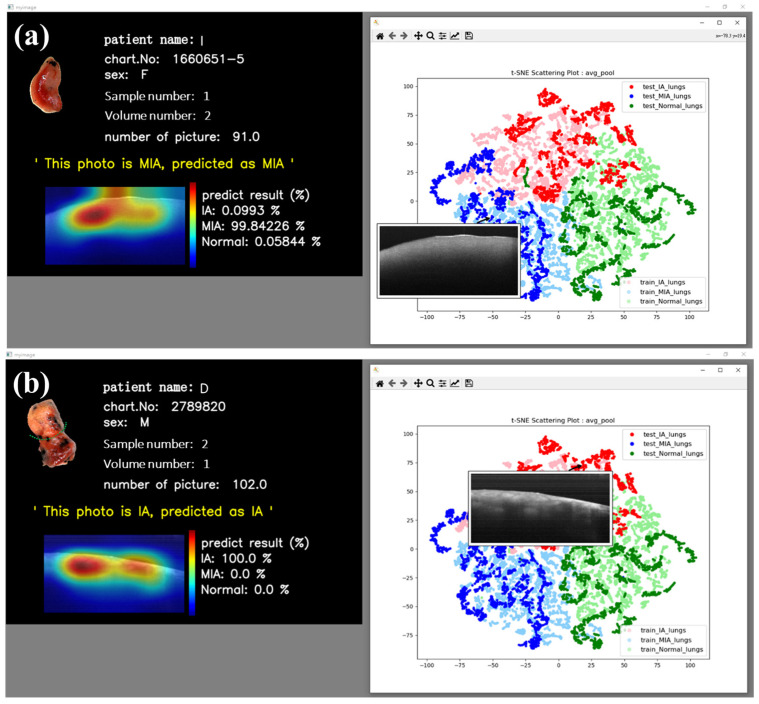
Deployment of interactive HMI. Results of (**a**) MIA and (**b**) IA show that our model performs well.

**Figure 8 cancers-15-05388-f008:**
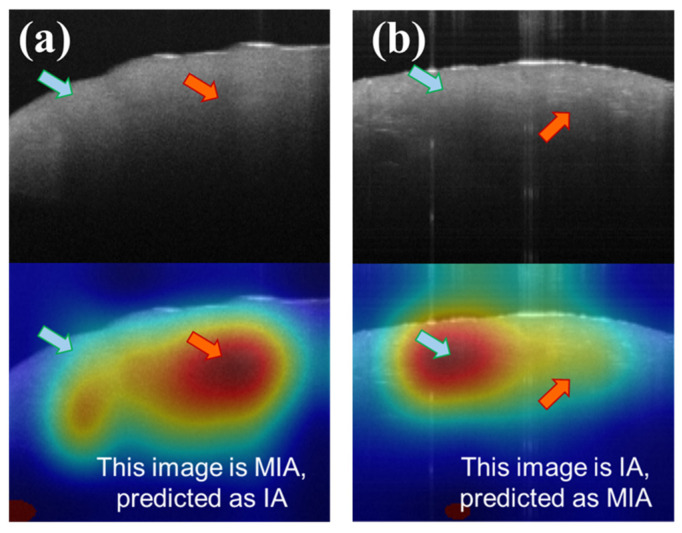
Image of neural network misclassification. (**a**) shows MIA being misjudged as IA, and (**b**) shows IA being misjudged as MIA. Red arrows mark the IA feature, while blue arrows point to the MIA feature.

**Figure 9 cancers-15-05388-f009:**
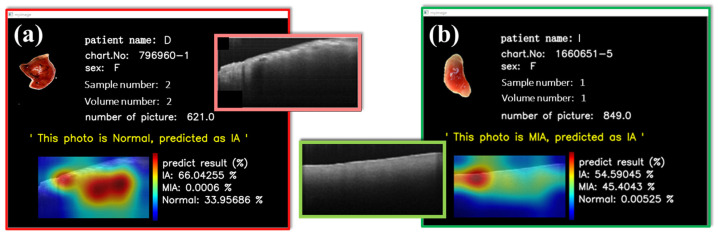
Misjudgments on interactive HMI. The model predicts three probabilities of MIA, IA, and NOR; the final prediction is the largest. (**a**) NOR from an IA patient erroneously categorized as IA. (**b**) Misclassification between IA and MIA. OCT-AI system provided the probabilities of tumor categories.

**Table 1 cancers-15-05388-t001:** Recruitment information.

Patient	Specimens (Tumor/Normal)		OCT Volumes		Permanent Diagnosis	Frozen Section	Data Splitting	
A	4		5		IA *	IA	Training	
	2		4				Training	
B	3		3				Training	
	2		2				Training	
C	3		3				Training	
	2		2				Training	
D	2		2				Testing	
	2		2				Testing	
E	2		2				Training	


F	2		4				Testing	


G	2		3				Training	


H	1		1		MIA **	IA	Training	
	1		1				Training	
I	2		4			MIA	Testing	
	1		2				Training	
J	3		3			AIS ***	Training	
	2		2				Training	
K	1		1			AIS	Training	
	1		1				Testing	
L	5		9			IA	Training	
	2		2				Training	

* IA, invasive adenocarcinoma; ** MIA, minimally invasive adenocarcinoma; *** AIS, adenocarcinoma in situ.

**Table 2 cancers-15-05388-t002:** Data separation. The columns from left to right are diagnosis type, OCT training data, and test data. The brackets are the number of C-scan groups, and the left side of the brackets is the total number of B-scans.

Diagnosis	Training		Testing	
	OCT Volumes	OCT Frames	OCT Volumes	OCT Frames
IA	16	12,120	6	3569
MIA	14	12,013	4	3069
NOR ^1^	14	13,225	4	3349
Total	44	37,358	14	9987

^1^ NOR, normal lung tissue.

## Data Availability

The data underlying the results presented in this paper are not publicly available according to the protection of human research participants.

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
