# Peer review of "Rapid On-Site AI-Assisted Grading for Lung Surgery Based on Optical Coherence Tomography"

_cancers, 2023, doi:10.3390/cancers15225388_

Round 1

Reviewer 1 Report

Comments and Suggestions for Authors

Liu et al. reported the Rapid on-site AI-assisted grading for lung surgery based on optical coherence tomography. This is an attempt to apply AI tools in making decisions on grading lung cancer surgery. Following minor revision is required before the acceptance:-

1. The introduction section needs improvement in terms of the application of AI tools in the concerned study.

2. Plagiarism detected from the author's previously published paper, "Hung‐Chang Liu, Miao‐Hui Lin, Ching‐Heng Ting, Yi‐Min Wang, Chia‐Wei Sun. "Intraoperative application of optical coherence tomography for lung tumor", Journal of Biophotonics, 2023. The plagiarism needs to be rectified in the revision.

3. Authors can also perform other AI methodologies like CNN, ANN, and SVM and compare the obtained results with these methodologies.

Author Response

We sincerely appreciate the reviewer's valuable comments and hints that have been very helpful in improving our revised manuscript. As a result, the revision is polished as carefully and thoughtfully as we can:

  1. We moved some paragraphs from Materials and Methods and Results to Discussion. In addition, we rearranged the order of paragraphs to make it easier for readers to understand. Paragraphs that have been moved or added are marked in red.
  2. We added a literature review mentioning OCT combination with AI techniques of cancer tissue in Introduction paragraph 3, and the reference rankings were redrafted for revision.
  3. To evaluate the performance of the model, we add a figure (Figure 5): learning curve and ROC curve.
  4. For the marking lesion of Figures 6 and 8 (OCT intensity and grad-CAM images), we added feature explanations in Results paragraph 5, and clearly explained misclassified images in Discussion paragraph 5.
  5. IA, MIA, and AIS were conceptually explained in medical terms and emphasized the importance of corrected image classification in Introduction paragraph 3.

We emphasized this study's motivation and core values to summarize this revised letter. The response to your suggestions and concerns are all addressed. In addition, we presented comparisons after modification and related descriptions in detail in this revision.

Reviewer 2 Report

Comments and Suggestions for Authors

IA, MIA, AIS should be conceptually explained in medical terms, and the factors that would influence the identification of the class of image.

Interpretations of each t-SNE image shown is appreciable.

Why is t-SNE dimensionality reduction chosen.

Comparison on choosing the dimensionality reduction technique should be elaborated .

parameters on choosing the Deep neural network and the selection of layers should be elaborated.

The ROC, loss, and accuracy metrics based on epoch should be analyzed and interpreted to show the trend of change in them with respect to epoch.

Since the RNN is adopted from the author's published article, a comparative analysis of the traditional network and the adopted network is to be done.

Author Response

(The authors gave the same response as above.)

Reviewer 3 Report

Comments and Suggestions for Authors

In this paper the authors develop a system based on deep learning, which performs tumor grading in real time.

The descriptions of the used methods and the results are good with all necessary technical details.

I recommend you to discuss in more detail the process of  mark lesions location. 

Author Response

(The authors gave the same response as above.)

Round 2

Reviewer 2 Report

Comments and Suggestions for Authors

No such further recommendation. Felt the updation is fine to move on publication.